# College Classroom Instructors Can Effectively Promote Standing among Students Provided with Standing Desks

**DOI:** 10.3390/ijerph18094464

**Published:** 2021-04-22

**Authors:** Matthew S. Chrisman, Robert Wright, William Purdy

**Affiliations:** School of Nursing and Health Studies, University of Missouri-Kansas City, Kansas City, MO 64108, USA; robert.wright@emory.edu (R.W.); wpurdy@ksu.edu (W.P.)

**Keywords:** standing desks, college, sedentary behavior, physical activity

## Abstract

Standing desks may reduce sedentary behaviors in college students. Students at one mid-size urban university in the Midwestern United States were randomized into intervention (*n* = 21) and control groups (*n* = 27) to assess standing time when given access to standing desks. The intervention group received visual and oral instructor prompts to stand, while the control received no prompts during a 50 min lecture. All students were provided with adjustable tabletop standing desks. ActivPAL accelerometers measured sitting and standing time. A brief survey assessed student preferences, including facilitators and barriers to standing. Mean standing time was greater in the intervention vs. control group (26 vs. 17 min, *p* = 0.023). Students tended to stand in the corners and edges of the room. Main facilitators for standing included to break up sitting, reduce back pain, and increase attention and focus; main barriers were not wanting to distract others or be the only one standing. In total, 87.5% of intervention group participants found five prompts to stand were adequate. Students increased standing time in class when provided with standing desks and instructor prompts to stand. Findings can inform the layout of classrooms and when and how to promote standing desks during lectures.

## 1. Introduction

College students spend almost 12 h per day and 68 h per week in sedentary behavior [1]. This is concerning given links between time spent sedentary, poor cognitive functioning, and mental distress [2]. With over 20 million students expected to be enrolled in United States (US) colleges by the fall of 2023 [3], reducing the effects of sedentary behavior on college students’ health is a significant public health issue.

One emerging solution to reducing sedentary behavior is via the use of standing desks in college classrooms. However, research using this strategy is sparse in the college setting. In adult workplace settings and primary and secondary schools, where the majority of research in this area has been conducted, sit-to-stand desk transitions have resulted in positive effects for reducing sedentary behavior among middle-aged office workers and school-aged youths [4]. In workplace settings, standing desks may be beneficial for improving both mental and physiological health outcomes [5], and an eight-week randomized intervention of 23 adult office workers found that standing desks significantly increased standing time by 72.9 min and reduced sitting time 80.2 min per 8 h workday, compared to 21 control group participants [6]. In K-12 school settings, standing desk interventions have found that standing time is 41 min per school day for children [7], with up to an hour of reduced sedentary time each day [4]. 

While effective at reducing sedentary behaviors in those populations, standing desk accommodations are not widely available in colleges today. From the limited literature available, students and instructors show enthusiasm for incorporating standing desks into classrooms, with data indicating 75% of students and 85% of instructors would be in favor of introducing them into their classes [8]. Moreover, task completion and performance were unaffected by whether university students stood or sat, and students exhibited greater interest, alertness, and engagement in tasks while standing [9]. Other preliminary research indicated college students in a classroom fitted with standing desks stood up to 7.2 min per hour [10]. There is a need to examine the best strategies to promote standing within the college classroom setting [11].

A potential strategy for promoting standing within classrooms is via instructor encouragement or prompts. One study of instructor-led activity breaks found that students used standing desks during class enjoyed activity cues from the instructor; however, the authors reported their classroom was set up for students to be highly active even without standing desks [12]. The use of visual prompts such as posters and table plaques has increased standing and reduced sitting time in college students in study areas and school libraries [13]. Evidence also shows the motivational effect of prompts in workplace office environments increased standing transitions over a sustained period of time [14]. No studies have examined instructor encouragement and visual cues to promote standing in a traditional lecture-style college classroom incorporated with standing desks, and none have examined the ideal number of prompts that might persuade a student to stand. Furthermore, there is a need to better understand facilitators and barriers to using standing desks in classrooms.

Studies of standing desks report increases in executive function and working memory among secondary students [15], and no negative classroom effects related to learning outcomes [16,17]. No studies examining academic outcomes in college students using standing desks have been conducted, and research is needed in order to inform standing desk interventions in this population. Therefore, the purposes of this single-masked, randomized intervention study were: (1) Determine amounts of sitting and standing time in college students when given access to a standing desk and being provided with visual and oral prompts to promote standing by college classroom instructors; (2) assess the acceptability of five visual and oral prompts to stand; and (3) examine facilitators and barriers to using standing desks, including any group differences that exist among college students. 

## 2. Materials and Methods

### 2.1. Sample and Recruitment

The Institutional Review Board at (blinded #19-012) approved all study procedures. The study was exempt, and informed consent was not collected; rather, consent was implied with participation in the study. Procedures occurred at this mid-size University located in an urban setting in the Midwestern US. The goal was to obtain 60 participants—30 in the control group and 30 in the intervention group. This desired sample size was limited due to financial and resource constraints, as well as the size of the available classroom. To be included, students had to be ≥18 years old and currently enrolled at the University. Exclusion criteria were not speaking English, being allergic to adhesives, and/or having an injury or medical condition that would hinder the ability to stand in class. Participants were provided with a $20 electronic gift card to a national retailer for their participation.

### 2.2. Data Collection and Intervention

The study was single masked as participants were unaware of their group assignment. Participants were randomly assigned to the intervention (instructor-provided visual and oral prompts to stand) or control group (no prompts), then provided with a date and time for the study. Study procedures occurred in the evening to avoid conflicts with regularly scheduled classes. Students arrived at a classroom that was outfitted with 15 tables, upon which rested two adjustable standing desks (Desk Riser 28X, Longmont, CO, USA) per table. These desks are adjustable from a height of 3 to 14 inches above the desk, with three separate height settings. Non-adjustable stools (Learniture, Cincinnati, OH, USA) were 24 inches high and available behind each desk. There were four rows of tables with an aisle between each. Students were randomly assigned to a desk as they entered the room (see Appendix A) and told they could sit or stand, whichever they were comfortable with. They were provided with instructions for attaching an ActivPAL device (see below) one-third of the way down the mid-axial line of their right thigh. Hypafix adhesive was used to apply the ActivPAL to the thigh and Tegaderm was used to secure the device in place. Two trained research assistants ensured students attached the devices correctly and made sure the devices were upright. Students were allowed to adjust their desk to find a comfortable height prior to the commencement of the study. The desks used in this study were found to be acceptable by college students in a previous study [11]. 

ActivPALs are a small, thigh-worn activity monitor designed to measure sitting and standing time [18]. They have demonstrated reliability and validity in measuring sedentary behaviors in college students and young adults [19]. Prior to the study, the ActivPALs were pretested for 1, 5, and 10 min bouts to ensure accuracy in sitting, standing, and sit-to-stand transitions; no issues were discovered. ActivPALs in this study assessed sitting and standing minutes and independent bouts, number of sit-to-stand transitions, and predicted MET hours expended. 

A health sciences instructor provided a 50 min lecture on a health topic unrelated to physical activity and sedentary behavior. The same instructor provided the lecture separately to both the intervention and control groups to maintain intervention fidelity. Students were instructed to pay attention as if it were a normal class lecture and were allowed to adjust the desks if desired. 

The intervention group received five visual and oral prompts to stand, which were inserted into the upper right corner of approximately every 5–6 PowerPoint lecture slides. The visual prompts were developed by the research team and included a small, animated image of a person standing at a standing desk with the words “Don’t forget to stand” across it and were accompanied by a brief animation to highlight them. Pretesting of the logos and their placement with two college students revealed that the logo at the top right corner was visible but not obtrusive and was preferred over the bottom corners or elsewhere on the slides. The oral prompt included the statement “As a reminder, don’t forget to stand”, which was delivered at the end of discussing slide content and prior to moving to the subsequent slide. The control group received the same lecture minus any visual or oral prompts to stand. Upon completion of the lecture, students were given a brief questionnaire to complete (see below). A digital scale (GreaterGoods, St. Louis, MO, USA) was available in the classroom for participants to record their weight. 

### 2.3. Questionnaire

The questionnaire instrument used in this study consisted of 13 questions and was a modified version of a questionnaire developed in a previous study [11] (see Appendix A). Five questions assessed demographics (age, gender, grade, race/ethnicity, and self-reported height, which was used to calculate BMI along with weight measured by digital scale in the classroom); four questions assessed if students stood at the desks (yes or no), reasons for standing or not (list of items with select all that apply), and barriers to standing (list of item with select all that apply); the final four questions assessed the optimal amount of time to stand in class (how many minutes per 60 min class), whether the prompts to stand were too many, just right, not enough, or did not see any (control group did not receive this question), and weekly self-reported sitting time in hours and minutes per day [20], which could be a confounding variable for standing time. The questionnaire was completed in person using pen and paper, took three to five minutes to complete, and was pretested by a convenience sample of three college students prior to the study commencement to ensure face validity. All questions were understood, and wording suggestions included: adding “In your opinion” to the question “what would be the optimal amount of time to stand in class”; adding “makes me feel more accountable in class” as a response option for a reason to stand; and adding “cultural or religious reasons” as a barrier to standing response option. 

### 2.4. Statistical Analyses

Analyses were conducted in SPSS version 26.0 (IBM Corporation, Armonk, NY, USA) using frequencies, proportions, and Chi-square tests to compare the intervention and control groups, as well as the demographic groups. Significance was set at *p* < 0.05, with Fisher’s exact test used for small sample size comparisons (*n* < 30) when appropriate. 

## 3. Results

A total of 106 students were recruited from the University over a four-week period to participate via flyers posted around campus, announcements in multiple health sciences classes, and word of mouth. There were 27 students randomized to the control group and 41 to the intervention group, representing a 64% response rate for those recruited. Reasons for not attending the study were not collected. Twenty students’ ActivPAL data were not captured correctly, and only their questionnaire data are reported. Upon contacting PAL Technologies, Ltd. (Glasgow, UK), the problem with the ActivPAL data capture was related to a software update that had not been completed, and a second intervention group had to be recruited to obtain siting and standing data. Table 1 includes the demographic information for those with ActivPAL data. Compared to the university demographics, study participants overall had a higher proportion of female students (90% in the study compared to 55% in the university) and were younger (average age of 21.5 years compared to 25 for the university) [21]. The study sample also included a higher proportion of Hispanic students (17% compared to 7%), and freshmen or seniors (38% and 40% compared to 9% and 16%, respectively) than the university as a whole. 

### 3.1. Standing and Sitting Time

All participants successfully attached and wore the ActivPALs for the entire 50 min lecture. A total of 7 (26%) participants in the control group and 19 (91%) in the intervention group reported standing during the intervention (Fisher’s exact test *p* < 0.001). Table 2 shows the mean standing and sitting time, along with sit-to-stand transitions and predicted MET hours expended, for those with ActivPAL data. Participants were more likely to sit for 30 min bouts in the control group compared to the intervention group (10 vs. 1, respectfully; *p* = 0.013 by Fisher’s Exact test). 

Students in the control group had 47 independent standing bouts (Table 2) which included 32 standing bouts of 0–10 min, 8 bouts of 11–20 min, and 3 bouts of 20–30 min. There were also four bouts greater than 30 min. Students in the intervention group also had 47 independent standing bouts which included 32 bouts of 0–10 min, 7 bouts of 11–20 min, 3 bouts of 20–30 min, and 5 bouts greater than 30 min. The number of bouts did not differ for the control versus intervention group on any of the durations. Figure 1 shows a heat map where students were standing in the classroom and for how long. For both groups, there was a slight tendency for students to stand more and longer on the edges and corners, with a shorter standing duration in the middle of the classroom. 

### 3.2. Reasons for or against Standing

Table 3 shows reasons for and against standing for the entire sample, and for the intervention and control groups separately. The most frequently reported reasons for standing included to break up sitting time (*n* = 31, 45.6%), reduce back pain (*n* = 28, 41.2%), and increase attention and focus (*n* = 26, 38.2%). All of the reasons to stand (except prefer standing to sitting, and standing helps learning) were reported significantly more frequently by the intervention group compared to the control. 

In terms of demographic group differences on reasons to stand, males were more likely than females to report standing as a result of seeing others standing (*X*[2] = 8.6; Fisher’s exact *p* = 0.015). Asians and Caucasians were more likely than the other race/ethnicity categories to report standing to break up sitting time (*X*[2] = 11.6; *p* = 0.009); and Asians were more likely than other race/ethnicity categories to report that standing makes them feel more accountable in class (*X*[2] = 8.8; *p* = 0.031). No group differences were found by grade level or BMI category. Other reasons students reported for standing included: feeling fidgety, standing to increase blood flow because “my lower half was feeling numb”, to help focus, and to reduce neck pain. 

The most frequent reasons against standing included that it would block others’ view (*n* = 15, 22.1%), would distract others (*n* = 12, 17.6%), and being too tired (*n* = 9, 13.2%). All of the reasons against standing were more frequently reported by the control group compared to the intervention group, with three of them more significantly reported (standing would block others’ view, standing would distract others, and no one else is standing). 

In terms of group differences against standing, freshman and sophomore students were more likely to report being too tired than all other grade levels (*X*[2] = 9.9; *p*-value = 0.042). There were no differences by gender, BMI category, or race/ethnicity. 

### 3.3. Barriers to Standing

Table 4 shows the most frequently reported barriers to standing by the overall sample and for the intervention and control groups. The most frequent barriers reported were not wanting to distract others, being tired, and not wanting to be the only one standing. Other barriers reported included: being in another’s way, not wanting to block others’ view (*n* = 2), and not being able to see around someone (*n* = 2). 

In terms of group differences, only one barrier was significantly different between the intervention and control groups (being unable to stand due to injury or health reasons, *p* = 0.009). Senior and sophomore students were more likely to report not wanting to invade their classmates’ personal space (*X*[2] = 11.5; *p* = 0.042). White students more frequently reported no one else standing as a barrier (*X*[2] = 14.4; *p* = 0.002), and Black students were more likely to report not knowing how to adjust the standing desk to be able to stand (*X*[2] = 8.2; *p* = 0.041) compared to the other race/ethnicities. No differences were found by gender or BMI category. 

### 3.4. Optimal Duration and Prompts to Stand

Students reported a mean optimal standing time of 23.8 min (SD 10.14) per 60 min class, with a range from 1 to 60 min (median = 20 min). It is interesting to note that the intervention group reported a greater amount of optimal standing time (25.5 min, SD = 10.9) compared to the control group (21.3 min, SD = 8.5), although it was not significantly different (*p* = 0.098). Further, females (24.5 min, SD 9.9) reported a larger optimal standing time than males (18.9 min, SD = 10.7), although this did not differ significantly either (*p* = 0.143). No meaningful patterns emerged among preferred optimal standing time by grade, BMI category, or race/ethnicity. 

For the intervention group only, students were asked to rate the amount of prompts they received. A total of 35 (87.5%) reported the 5 prompts to stand were just right, while 2 (5%) reported they were too many; 2 (5%) reported they were not enough; and 1 (2.5%) reported they did not receive any prompts. 

## 4. Discussion

This study examined whether instructor-provided visual and oral prompts would increase standing time in college students in a classroom with adjustable standing desks, and facilitators and barriers to using them. Given that all students wore the ActivPALs for the entire study period, these devices were a feasible measurement tool for this short-duration, randomized intervention study. Results showed that when all students were given access to tabletop adjustable standing desks, they used them for an average of almost 17 min per 50 min lecture, and when provided with the visual and oral prompts, they engaged in more 9 more minutes of standing compared to having no prompts. The five oral and visual prompts were acceptable.

College students may sit for 6.18 h per weekday during lectures and other school-related activities that are considered sedentary [22]. Increasing standing time could improve their health [23], and our study showed that access to standing desks provides a feasible option to break up sitting time in college lectures. In fact, only 11 (23%) participants in this study engaged in a bout of sitting for at least 30 min. These findings may have broader implications since interrupting prolonged sitting has been associated with greater cognitive performance [24]. Students who stand in class may exhibit greater cognition, potentially improving academic performance. 

The mean standing time for the intervention and control groups was greater than a previous intervention which found college students stood for 7 min per hour when given access to a classroom standing desk [10]. Further, intervention participants in this study stood an average of 9.4 more minutes than the control group, which could amount upwards of 300 fewer minutes of sedentary time over an entire semester for college classes that meet two hours per week. Providing students with the option to stand in class via adjustable standing desk increased standing even in the absence of instructor prompts; the addition of visual and oral prompts was effective at further increasing standing. Taken as a whole, these data can inform future research regarding when to incorporate visual or oral prompts to stand in classrooms. 

In both groups, 68% of standing bouts lasted for 0–10 min, with most of those bouts occurring for 5 min or less. One previous study of 24 college students also found a high proportion of standing bouts lasted 10 min or less [25]. The total number of standing bouts was the same in each group in this study, which supports existing evidence [10]. Students sitting for more than 30 min during the lecture accounted for over 30% of the control group compared to only 5% of the intervention group, indicating that visual and oral prompts were effective methods for reducing sedentary behavior in college classrooms. Other evidence supports visual cues and reminders from instructors as being positively associated with students engaging in less sedentary behavior during class [12,13]. 

The highest concentrations of both groups’ total standing time occurred at desks placed along the edges and corners of the classroom. These findings suggest that students may find it a barrier to stand during class if they are placed toward the middle of the classroom. Conversely, these findings are consistent with social norms facilitators, indicating that seeing other students stand during lectures encourages standing [10]. An encouraging finding is that students stood in the front row. Others have placed adjustable standing desks in the back of college classrooms [25], presumably to limit distractions or blocking others’ view, and findings in our study indicate that standing desks may be effective on all edges of the classroom. 

The top three reported reasons for using a standing desk were to break up sitting time (*n* = 31), to reduce back pain (*n* = 28), and to increase attention and ability to focus (*n* = 26). Knowing the reasons students stand could perhaps be used in future messages and prompts to promote standing in college classrooms, particularly since others have found that visual decisional cues can increase standing in college students [13]. Students in this study overwhelmingly indicated that the five visual and oral prompts were adequate. 

Other studies have found that students reported using standing desks led to improvements in attention, physical health, and focus while decreasing restlessness during class time [8,10]. A small number of participants in this study (*n* = 7) reported that standing could help to improve their ability to learn in class. These results indicate that students likely feel more engaged, less tired, and less bored while using standing desks in class, which may lead to better academic performance. Given that less than 3% of students have reported using standing desks in college [8], more studies are needed to examine this outcome.

One interesting finding is that the intervention group was more likely to support reasons for using standing desks than the control group, indicating that perhaps the simple act of standing makes it more apparent why one should engage in that behavior. Similarly, the control group generally was more likely to support reasons against standing compared to the intervention group, indicating that perhaps the act of sitting makes it more apparent why one should engage in that behavior. 

Students reported that the major barriers to standing were not wanting to distract others, being tired, and not wanting to be the only one standing. This supports the finding that the highest concentration of standing occurred at the edges of the classroom, which might be less distracting to others. Furthermore, these findings indicate that students may be encouraged to stand if others are standing as well and emphasize that standing is a social behavior in the college environment. Others have found that college instructors perceive standing desks would best be located in the back or edges of classroom rows of seats [8], and consideration of this evidence would be useful when designing future standing-desk-friendly classrooms. 

The group differences found among reasons for and against standing, and barriers to using standing desks, provide additional support in the literature in regard to gender differences [26] and age/grade differences [27], which should be considered in future interventions. In particular, gender should be targeted in terms of social interventions, and research should examine why students in younger grades report being more tired. 

Students with access to standing desks reported they would stand between 25% and 50% of class time [8]. This could lead to an increase between 12.5 min and 25 min in standing time during a 50 min lecture. In this study, students reported a mean standing time of 24 min per 60 min class would be optimal. Research is needed to examine if this amount of standing is feasible in college classrooms, and whether it would affect student academic performance. 

Interrupting sedentary bouts every 10–20 min was associated with better academic achievement in college students [28], which is important given that students in this study reported the five oral and visual prompts were adequate. Spacing these prompts out equally would provide an interruption approximately every 10 min of class time. These brief interruptions are simple and easy, requiring little cognitive energy, which is ideal for contributing to sustained behavior changes [29]. Further, college students experience sleepiness after just 15 min of class time, and uninterrupted sitting time is associated with physical discomfort [30]. Interrupting sedentary bouts via standing at standing desks in classrooms could contribute to maintaining alertness, and future studies should examine if they also contribute to reducing discomfort. Further research is needed to confirm if the optimal number of prompts is five in other college settings, including longer class periods. 

### Strengths and Limitations

Strengths of this study include the randomization of participants, inclusion of a control group, the use of a validated device for measuring sitting and standing, and the real-world nature of the intervention. There were a few limitations. The sample size was relatively small and not fully representative of the University student population. In particular, the low percentage of male participants indicates an area for future research on standing desks in college students. This study occurred in a mid-sized, urban Midwestern university, and findings may not be generalizable to other settings. Further, the ActivPALs were only used to measure siting and standing behavior for 50 min, which is a shorter duration than validity studies using this device. Our lab pretested this duration, which more likely mimics a standard classroom lecture, and found it feasible and valid.

## 5. Conclusions

Based on the results of this randomized intervention study, instructor-provided visual and oral prompts are effective at increasing standing time and reducing sitting time in college students given access to adjustable classroom standing desks. Five visual and oral prompts were acceptable. Future interventions and classroom activities incorporating standing or activity breaks should target short bouts which appear to be more favored by college students. Students tended to stand more in the classroom edges and corners; these results could contribute to the design and layout of activity-friendly college classrooms. Further, the study provides evidence for using ActivPALs to assess standing and sedentary behaviors in shorter durations, such as a classroom lectures. The devices would benefit from more rigorous testing of this method, particularly as it relates to these behaviors in college students.

Based on the results of the current study, college students may prefer using standing desks to reduce sitting time, reduce back pain, and/or increase attention in classrooms, among other things. Students suggested the optimal time spent standing during class was almost half of a 60 min class, and it would be interesting to see academic and health effects of this amount of standing.

Future research is needed to examine the long-term feasibility and acceptability of using standing desks in a college classroom, as well as instructor perspectives for using and promoting standing desks. While college instructors have demonstrated support for having standing desks [8], there is a lack of studies examining their need as it relates to feasibility of using these desks in a variety of educational and curriculum settings. Research should examine whether exposure to standing desks in a college classroom could translate into more frequent standing in other domains of college student life, including at home and work. It may also be worth examining ideal classroom layouts to maximize standing time.

As it relates to the topic of this Special Issue, we believe the evidence in this study and elsewhere indicates a need to develop sedentary recommendations, or limits, for what is acceptable in terms of sitting or expending ≤1.5 METs. Physical activity and sedentary behavior have often been presented as opposite sides of the same spectrum, yet we would argue that they belong on individual spectrums with unique health concerns. From our results, it is clear that sedentary behavior in young adults can be reduced with the provision of a standing desk in a college classroom, and further reduced with instructor prompts to use the desks for standing. Findings also offer participants’ suggestions for the optimal amount of time to stand during classroom lectures, which can provide a starting point for interventions.

## Figures and Tables

**Figure 1 ijerph-18-04464-f001:**
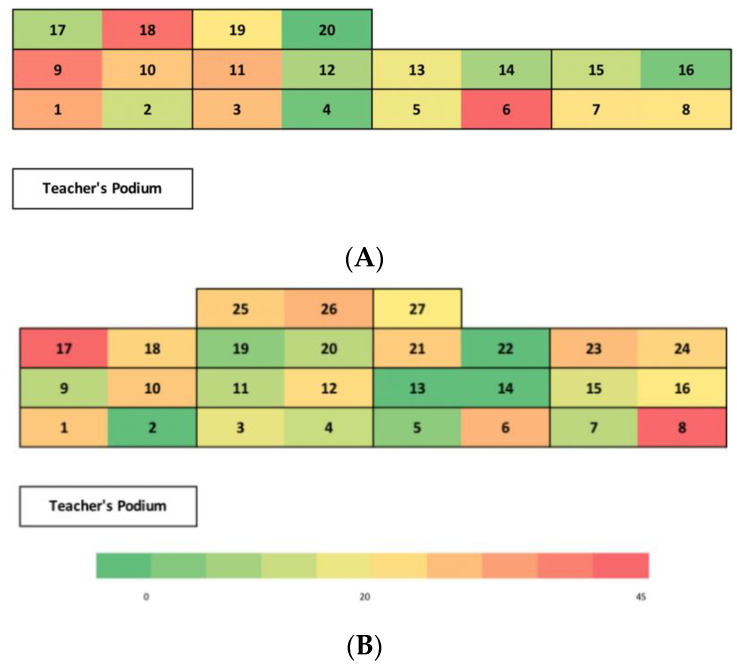
Heat map showing total number of minutes of standing in the intervention ((**A**), **top**) and control ((**B**), **bottom**) groups, and where the standing desks were located in the classroom. The color scale indicates the amount of standing, with less standing in the green, and the most standing in the red. The numbers on the bottom color scale indicate the number of minutes stood.

**Table 1 ijerph-18-04464-t001:** Demographic characteristics of the study participants with ActivPAL data.

Variable	Category	Overall Sample (*n* = 48)	Intervention (*n* = 21)	Control (*n* = 27)	* *p*-Value
Age in years, mean (SD)		21.5 (4.9)	20.6 (5.3)	22.2 (4.5)	** 0.70
Gender, *n* (%)	Male	5 (10%)	1 (5%)	4 (15%)	0.37
	Female	43 (90%)	20 (95%)	23 (85%)	
Grade, *n* (%)	Freshman	18 (38%)	14 (67%)	4 (15%)	0.001
	Sophomore	2 (4%)	0 (0%)	2 (7%)	
	Junior	7 (15%)	1 (5%)	6 (22%)	
	Senior	19 (40%)	5 (24%)	14 (52%)	
	Graduate	1 (2%)	0 (0%)	1 (4%)	
	Other	1 (2%)	1 (5%)	0 (0%)	
Race, *n* (%)	White	24 (50%)	11 (52%)	13 (48%)	0.04
	Hispanic	8 (17%)	1 (5%)	7 (26%)	
	Black	8 (17%)	2 (10%)	6 (22%)	
	Asian	5 (10%)	4 (19%)	1 (4%)	
	Other	1 (2%)	1 (5%)	0 (0%)	
	Two or more	2 (4%)	2 (10%)	0 (0%)	
BMI, *n* (%) ^	Underweight	1 (2%)	0 (0%)	1 (4%)	0.63
	Normal	25 (52%)	13 (62%)	12 (44%)	
	Overweight	15 (31%)	5 (24%)	10 (37%)	
	Obese	7 (15%)	3 (14%)	4 (15%)	

* *p*-value from Fisher’s exact test between the intervention and control groups except where otherwise indicated; ** *p*-value from Mann–Whitney U test between the intervention and control group. ^ BMI cutpoints based on the BMI index from the Centers for Disease Control and Prevention.

**Table 2 ijerph-18-04464-t002:** Standing and sitting time of study participants during the 50 min lecture for those with ActivPAL data.

Variable	Intervention (*n* = 21)	Control (*n* = 27)	*p*-Value *
Sitting minutes, mean (SD)	18.5 (11.3)	27.7 (17.0)	0.028
Standing minutes, mean (SD)	26.3 (11.4)	17.0 (16.7)	0.023
Sit-to-stand transitions, mean (SD)	2.1 (1.5)	1.6 (1.2)	0.273
Standing bouts 0–5 min, *n* (%) ^	22 (47%)	29 (62%)	0.214
Standing bouts 6–10 min, *n* (%) ^	10 (21%)	3 (6%)	0.07
Standing bouts 11–15 min, *n* (%) ^	6 (13%)	3 (6%)	0.486
Standing bouts 16–20 min, *n* (%) ^	1 (2%)	5 (11%)	0.204
Standing bouts >20 min, *n* (%) ^	8 (17%)	7 (15%)	1.00
Sitting Bouts of 30 min, *n* (%)	1 (5%)	10 (37%)	0.013
MET hours, mean (SD)	1.01 (0.03)	0.98 (0.05)	0.047

* Mann–Whitney U test between intervention and control groups; ^ percentage of total standing bouts with Fisher’s exact *p*-value.

**Table 3 ijerph-18-04464-t003:** College students’ most frequently reported reasons for and against standing from most to least frequent, including the differences between the intervention and control groups.

Reason	Overall, *n* (%)	Intervention, *n* (%)	Control, *n* (%)	*p*-Value *
For standing				
To break up sitting time	31 (45.6)	26 (63.4)	5 (18.5)	<0.001
To reduce back pain	28 (41.2)	22 (53.7)	6 (22.2)	0.011
Increases attention/focus	26 (38.2)	22 (53.7)	4 (14.8)	0.001
Encouraged by instructor to stand	22 (32.4)	22 (54.7)	0 (0.0)	<0.001
To improve health	18 (26.5)	15 (22.1)	3 (11.1)	0.023
Helps reduce phone/laptop distractions	14 (20.6)	13 (31.7)	1 (2.1)	0.005
I prefer standing to sitting	12 (17.6)	9 (22.0)	3 (11.1)	0.33
Standing desks are new and cool	11 (16.2)	10 (24.4)	1 (2.1)	0.021
Makes me feel more accountable in class	11 (16.2)	10 (24.4)	1 (2.1)	0.021
Others are standing	10 (14.7)	10 (24.4)	0 (0.0)	0.004
Can see the instructor/front of room better	8 (11.8)	8 (19.5)	0 (0.0)	0.017
Standing helps me learn better in class	7 (10.3)	5 (12.2)	2 (7.4)	0.691
Against standing				
Standing would block others’ view	15 (22.1)	2 (4.9)	13 (48.1)	0.002
Standing would distract others	12 (17.6)	1 (2.4)	11 (40.7)	0.003
Too tired	9 (13.2)	4 (9.8)	5 (18.5)	1.00
I prefer sitting to standing	8 (11.8)	2 (4.9)	6 (22.2)	0.255
Standing feels awkward socially	7 (10.3)	1 (2.4)	6 (22.2)	0.103
No one else is standing	7 (10.3)	0 (0.0)	7 (25.9)	0.011
No encouragement to stand	5 (7.4)	0 (0.0)	5 (18.5)	0.052
Desk is too short or tall	4 (5.9)	0 (0.0)	4 (14.8)	0.113
Do not want to be seen standing	3 (4.4)	0 (0.0)	3 (11.1)	0.238
Unable to stand due to injury or other health reason(s)	2 (2.9)	1 (2.4)	1 (11.1)	1.00
Cultural or religious reasons	1 (1.3)	0 (0.0)	1 (11.1)	1.00
Unable to use laptop/phone while standing	0 (0.0)	__	__	__

* *p*-value using Fisher’s exact test.

**Table 4 ijerph-18-04464-t004:** Barriers to standing from most to least frequently reported by college students, including differences between intervention and control groups.

Barrier	Overall, *n* (%)	Intervention, *n* (%)	Control, *n* (%)	*p*-Value *
Don’t want to distract others	45 (66.2)	28 (68.3)	17 (63.0)	0.794
Being tired	43 (63.2)	27 (65.9)	16 (59.3)	0.615
Don’t want to be the only one standing	41 (60.3)	27 (65.9)	14 (51.9)	0.314
No one else is standing	39 (57.4)	27 (65.9)	12 (44.4)	0.132
Desk is too short or tall	25 (36.8)	15 (36.6)	10 (37.0)	1.00
Did not know how to adjust the desk to be able to stand	25 (36.8)	18 (43.9)	7 (25.9)	0.199
No encouragement to stand	24 (35.3)	16 (39.0)	8 (29.6)	0.452
Unable to stand due to injury or health reason(s)	23 (33.8)	19 (46.3)	4 (14.8)	0.009
Don’t want to invade neighbor’s space	22 (32.4)	15 (36.6)	7 (25.9)	0.433
It is not comfortable to stand	21 (30.9)	13 (31.7)	8 (29.6)	1.00
Desktop is too small	8 (11.8)	7 (17.1)	1 (11.1)	0.133
Cultural or religious reasons	4 (5.8)	4 (9.8)	0 (0.0)	0.146

* *p*-value using Fisher’s exact test.

## Data Availability

Data supporting reported results can be obtained by contacting the corresponding author.

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
