# Peer review of "College Classroom Instructors Can Effectively Promote Standing among Students Provided with Standing Desks"

_ijerph, 2021, doi:10.3390/ijerph18094464_

Round 1

Reviewer 1 Report

In this study of authors had multidimensional aims with a important variable (standing).

The Introduction was well written with adequate citations. Authors elaborated the importance of standing and current sedentary trends. They also analyzed the role of instructor in standing promotion.

The methods section was well described including inclusion criteria. In detailed data collection and Intervention were presented with special attention to the ActivPAL. The questionnaire that was used in this study was a modified version. Was that modified version previously validated? For face validity what was the method of choice and criteria to include 3 college students? What was the feedback of these students?

Results were clearly presented and tables and graphs are understandable, however it is not clear what was the method of testing the validity of ActivPAL, since it is one of the postulated aims.

In Discussion section authors stated that ActivPAL is accurate and valid measurement, again, the validation methods is not clear..?

Regarding the barriers authors stated that students reported the major barriers to standing were not wanting to distract others, being tired, and not wanting to be the only one standing. This is of great importance for improvement of better health habits.

Strengths and limitation were adequate.

Author Response

Thank you for the critical feedback. Attached please see a table of your comments with our responses, including where changes were made in the manuscript text!

Reviewer 2 Report

This study is interesting and in that it explores the use of standing desks in the college environment – an under-served area. The main finding is that desk provision did lead to standing time – and more so in the intervention (prompts provided) than the control (no prompts) group. Unfortunately, I feel that there are some significant issues with the manuscript, particularly around methods, furthermore the sample size is low, and the aims of the study are very unclear at the outset. I hope that the notes below are of use for a future submission.

Title: I think this is unclear – could it be re-worded to highlight the key purpose of the study? Probably best to drop ‘US’ reference as it reflects only one college site.

Intro: define K-12 setting for non-US audience

The introduction needs to be re-worked to make clear what the gap in evidence is, given this study specifically explores use of prompts in facilitating use of stand-up desks.  The evidence is contained in the introduction, but could be better structured to justify the inclusion of the various purposes listed.

I found the rationale for use of ActivPALS (penultimate para in intro) slightly adjunct to the preceding rationale in the intro – particularly as the first aim is around testing validity of the devices. Suggest integrate a need for objective monitoring earlier in the intro. Link to the dearth of evidence? I am also left slightly confused at this point but the four purposes stated. I suggest that these are clearly defined as overarching purpose and specific aims. There seems to be an awful lot here and it is not clear what the key purpose actually is, nor how this will be achieved.  

Methods:

Please state study design at the outset

Were students informed of the research purpose at recruitment? Or was this masked? Is there a rationale for sample size? Line 82 states that 106 students were recruited, but then later on (line 154) states that 27 and 41 students were randomized to control and intervention – please explain?

Whilst you state the ethical approval was granted, please describe processes for obtaining informed consent.

It would be useful to understand what information students were given about the desks, as they entered the room? Were they asked to choose whether to sit or stand? Would desks have needed to be adjusted manually to facilitate sitting down?

2.3 Questionnaire repeats some info on demographics presented in previous section, suggest removing some of the information in section 2.2 line 131

Please clarify which questions were open and which had options provided – and indicate what those options were – and how they were derived in context of extant literature/theory. Might be helpful to include a copy of the Q as a supplementary file?

Was the impact of peer-influence considered, given students were together in the classroom for the intervention group? Was this assessed in any way?

Line 138 – suggest that BMI procedures are fully documented, perhaps not as part of the questionnaire description.  Please provide details of procedures and scales used and BMI cut-points applied.

At end of methods I do not know how aim 1 is being considered – how were the ActivePALs validated? This is not considered in the results either.  Does the aim need to be removed, perhaps?

Results:

Line 154 Why specifically were 27 recruited to control and 41 to intervention group, rather than equal groups?

Table 1 – please state (in legend?) what the p values reflect – e.g. differences in frequencies between IV and control.

Figure 1 – whilst interesting, I am not sure this figure adds to the results, or interpretation of results, considering students did not (I don’t think?) have a opportunity to move around the room – and there is no clear discernible pattern? If the figure is to be retained, suggest reconsidering the colour scheme - it seems counter-intuitive – perhaps green might denote the positive health behaviour? And also suggest clearly labelling as a) and b) or similar and denoting which is intervention and which control in sub-title.

Discussion – this section provided a good overview of findings and to my mind, is a much stronger section of the paper.  

Author Response

Thank you for the critical feedback. Attached please see a table of your comments with our responses, including where changes were made in the manuscript text!

Comment

Response

This study is interesting and in that it explores the use of standing desks in the college environment – an under-served area. The main finding is that desk provision did lead to standing time – and more so in the intervention (prompts provided) than the control (no prompts) group. Unfortunately, I feel that there are some significant issues with the manuscript, particularly around methods, furthermore the sample size is low, and the aims of the study are very unclear at the outset. I hope that the notes below are of use for a future submission.

Thank you for your encouragement and critical feedback. We have addressed your concerns and those of the other reviewers as best as we could and appreciate the time you have taken to review our work.

Title: I think this is unclear – could it be re-worded to highlight the key purpose of the study? Probably best to drop ‘US’ reference as it reflects only one college site.

We have removed “in the US” from the title and edited it to highlight the key purpose. Thank you for the suggestion.

Intro: define K-12 setting for non-US audience

We have changed K-12 to “primary and secondary” schools. Thank you for suggesting this.

The introduction needs to be re-worked to make clear what the gap in evidence is, given this study specifically explores use of prompts in facilitating use of stand-up desks.  The evidence is contained in the introduction, but could be better structured to justify the inclusion of the various purposes listed.

We removed Aim 1, and edited the introduction to highlight another reviewer’s suggestion to discuss the effects of standing while learning. This has made the gap more clear. Thank you for the suggestion.

I found the rationale for use of ActivPALS (penultimate para in intro) slightly adjunct to the preceding rationale in the intro – particularly as the first aim is around testing validity of the devices. Suggest integrate a need for objective monitoring earlier in the intro. Link to the dearth of evidence? I am also left slightly confused at this point but the four purposes stated. I suggest that these are clearly defined as overarching purpose and specific aims. There seems to be an awful lot here and it is not clear what the key purpose actually is, nor how this will be achieved.

We have removed Aim 1 from the manuscript to focus the attention on the findings of the intervention. Thank you for noting this. We also edited the introduction to focus on the lack of studies looking at academic effects of standing, per another reviewers’ suggestion.

Please state study design at the outset

We have indicated the design in line 64.

Were students informed of the research purpose at recruitment? Or was this masked? Is there a rationale for sample size? Line 82 states that 106 students were recruited, but then later on (line 154) states that 27 and 41 students were randomized to control and intervention – please explain?

Students were not informed of the research purpose as that was part of the masking. The recruitment goal was 30 per group, which was based on funding and resource limitations. We have added this to the text in lines 81-82 to be more clear.

Our final sample included 27 in the control (three participant no-shows) and 41 in the intervention which was due to the ActivPAL devices not capturing data in the first data collection attempt. We had to collect the intervention data a second time, giving us data for 21 participants with ActivPAL data, but 41 with survey data. We have made this more clear in lines 137-138.

Whilst you state the ethical approval was granted, please describe processes for obtaining informed consent.

We added info on consent to lines 79-80. Briefly, the study was exempt and consent was not collected, but rather, was implied with participation.

It would be useful to understand what information students were given about the desks, as they entered the room? Were they asked to choose whether to sit or stand? Would desks have needed to be adjusted manually to facilitate sitting down?

We added info regarding what students were told as they entered the room. We mentioned that students were allowed to adjust the desks. The desks were randomly raised or lowered prior to the study to examine if that affected sitting and standing; however, it had no effect on outcomes and the data are not reported here. Students were allowed to adjust the desk manually to be comfortable for their choice to sit or stand.

2.3 Questionnaire repeats some info on demographics presented in previous section, suggest removing some of the information in section 2.2 line 131

We have removed the redundant info from line 118. Thank you for the suggestion.

Please clarify which questions were open and which had options provided – and indicate what those options were – and how they were derived in context of extant literature/theory. Might be helpful to include a copy of the Q as a supplementary file?

We have added info on the answer response options. We would be happy to include the questionnaire as a supplemental file.

Was the impact of peer-influence considered, given students were together in the classroom for the intervention group? Was this assessed in any way?

We assessed the impact of peers in the questions on reasons for standing or not, and barriers to standing, which included response options such as “others are standing” (reason to stand) or “standing would distract others” (barrier to standing). We also assessed where students were standing in the heatmap in Figure 1, which helps indicate where students stood in relation to where others were located. Peer influence would be an interesting area for future research, particularly given that two of the most frequent reasons to not stand were that it would distract others or block their view. 

Line 138 – suggest that BMI procedures are fully documented, perhaps not as part of the questionnaire description.  Please provide details of procedures and scales used and BMI cut-points applied.

We added a footnote in Table  1 to indicate where the BMI cutpoints came from. Line 119 indicates the scale used. Students could weigh themselves and self report their weight on the survey instrument.

At end of methods I do not know how aim 1 is being considered – how were the ActivePALs validated? This is not considered in the results either.  Does the aim need to be removed, perhaps?

We have removed Aim 1 from the manuscript. Thank you for the suggestion.  

Line 154 Why specifically were 27 recruited to control and 41 to intervention group, rather than equal groups?

As noted above, and in line 145, the initial intervention group’s (n=20) ActivPALs did not record data, so we had to recruit a second intervention group. The original plan was 30 students per group due to financial and resource limitations. Student participants not showing up led to different sized groups. We did not collect information from students who did not show up.

Table 1 – please state (in legend?) what the p values reflect – e.g. differences in frequencies between IV and control.

This was done. Thank you for the suggestion.

Figure 1 – whilst interesting, I am not sure this figure adds to the results, or interpretation of results, considering students did not (I don’t think?) have a opportunity to move around the room – and there is no clear discernible pattern? If the figure is to be retained, suggest reconsidering the colour scheme - it seems counter-intuitive – perhaps green might denote the positive health behaviour? And also suggest clearly labelling as a) and b) or similar and denoting which is intervention and which control in sub-title.

In the figure, the 5 desks that had red (the most standing) were all located in the edges of the classroom, which we discerned as a pattern. Students were free to move around if they desired. We have added an A and B as you suggested to make it more clear which group is which. This is also in the figure caption. Regarding the color scheme, those are the default colors provided in the ActivPAL software, which we are unaware of how to change. We do agree that perhaps more intuitive colors could enhance the ActivPAL data.

Discussion – this section provided a good overview of findings and to my mind, is a much stronger section of the paper.  

Thank you for the encouraging words!

Reviewer 3 Report

The sedenterine lifestyle is now a serious social problem in developed and developing countries. It is necessary to undertake actions aimed at activating the society to engage in physical activity. It is also important to stimulate frequent changes of posture while studying and working in the office, and to limit the sitting position. Therefore, I consider the research undertaken by the authors of the manuscript to be important and worth publishing. However, it should be remembered that working in a standing position does not solve the problem of a sedentary lifestyle, as it does not significantly increase the caloric expenditure. Taking into account the possible reduction of the negative impact of sitting on a human, one should strive to change the position frequently while studying and working in the office. Therefore, in the context of pro-health benefits, the use of frequent active breaks related to taking up physical exercises seems to be more important than practicing in a standing position.

Although I rate the manuscript highly, I have some comments for its authors:

In the introduction, we should point out the research showing the benefits of learning while standing and devote more attention to this issue.

In the "Materials and Methods" section, the ethics committee consent number to conduct the research should be provided.

It would be worthwhile to present an illustration showing the standing desks used in the research, which can also be a promotion of this type of equipment. This type of furniture is not known and used everywhere.

The part on statistics should be included in a separate subchapter, which could be called, for example, "Statistical Analysis"

In the "Results" section, please provide the reason why "Twenty students' ActivPAL data did not capture correctly and only their questionnaire data is reported".

In the group of respondents there was a significant majority of women (90%). Wasn't it necessary to give up men and limit the research to only women, and then repeat the research in the group of men? Given this imbalance, comparisons between men and women are questionable.

Also, comparisons between ethnic groups seem questionable given the relatively small number of respondents. I propose to consider giving up analyzing these dependencies.

A more detailed description of the heatmap legend should be considered. What does, for example, the color orange mean?

The notation of symbols of the p-value indicator, which is sometimes written with capital letters and sometimes with lower case letters, should be standardized. I suggest using lowercase letters.

Author Response

(The authors gave the same response as above.)
